# Pediatric Hepatocellular Carcinoma: A Review of Predisposing Conditions, Molecular Mechanisms, and Clinical Considerations

**DOI:** 10.3390/ijms26031252

**Published:** 2025-01-31

**Authors:** Elizabeth P. Young, Allison F. O’Neill, Arun A. Rangaswami

**Affiliations:** 1Department of Pediatrics, Division of Oncology, University of California San Francisco, San Francisco, CA 94158, USA; betsy.young@ucsf.edu; 2Department of Pediatric Oncology, Dana-Farber Cancer Institute, Boston Children’s Hospital and Harvard Medical School, Boston, MA 02215, USA; allison_oneill@dfci.harvard.edu

**Keywords:** hepatocellular carcinoma, pediatric HCC, pediatric liver tumors

## Abstract

Pediatric hepatocellular carcinoma (HCC) is a rare malignant liver tumor affecting children and adolescents and occurring either sporadically or in the context of underlying liver disease. In this review, we detail the epidemiology of pediatric HCC with a focus on predisposing factors including hepatic or systemic disease, genetic disorders, and familial cancer syndromes. We summarize existing research on the pathophysiology of pediatric HCC, including molecular mechanisms of oncogenesis, highlighting unique disease features differentiating pediatric HCC from adult HCC. We then survey the landscape of therapeutic options for pediatric HCC, including novel therapeutics. Lastly, we discuss the pathologic spectrum upon which pediatric HCC is postulated to exist, ranging from hepatoblastoma to HCC and including the hybrid entity hepatocellular neoplasm not otherwise specifed (HCN-NOS). In summary, we highlight the key clinical and molecular features of pediatric HCC that may inform future research and novel approaches to the clinical care of these patients.

## 1. Introduction

Pediatric hepatocellular carcinoma (HCC) is a rare malignant liver tumor affecting children and adolescents. While the majority of HCC cases are sporadic, arising de novo in a structurally and functionally normal liver, predisposing anatomic, metabolic, or genetic conditions have been reported to occur in approximately 10–50% of cases [1,2]. Historically, pediatric HCC has been studied in aggregate with pediatric hepatoblastoma (HB) in prospective therapeutic trials; as a result, it has been recognized that pediatric HCC has an inferior chemotherapy response rate, that complete resection is required for cure but often unattainable, and that outcomes are inferior. Over the last decade, multiple groups have published genomic work suggesting that pediatric HCC exists on a spectrum with HB and hepatocellular neoplasm not otherwise specified (HCN NOS), a hybrid entity with both HB and HCC features occurring in an intermediary age group. While informative for prognosis, few druggable targets have been identified. At present, the mainstay of therapy for pediatric HCC is derived from historic HB trials and relies upon a chemotherapeutic backbone. For the first time in history, pediatric HCC was studied in an international prospective trial, PHITT/AHEP1531/JPLT-4, which was powered to provide hopeful insights on patient and disease characteristics and treatment response as well as yield important biological information. Multiple single-institutional or small consortia trials are currently underway to study immunotherapeutics in pediatric HCC, mirroring the approach taken to the treatment of adult patients with HCC. The results of these studies, along with embedded correlatives, will undoubtedly inform novel approaches to treat this difficult-to-cure disease going forward.

## 2. Clinical Distinctions Between Hepatoblastoma and Pediatric HCC

The incidence of primary malignant pediatric hepatic tumors in the United States is approximately two cases per million children per year, accounting for 0.5 to 2% of all pediatric malignancies [3]. The two principal histological subtypes are hepatoblastoma (HB) and hepatocellular carcinoma (HCC). HB comprises 80% of all cases and predominantly affects children under 4 years of age. HCC occurs more frequently in adolescents and young adults but can occasionally occur in a young child. Hepatoblastoma is an embryonal tumor originating from fetal hepatocytes that tends to be chemosensitive and carry an excellent prognosis, with exceptions being for patients with metastatic disease or unfavorable biologic features. While most cases of HB are sporadic, associations with germline predisposition syndromes such as Beckwith–Wiedemann Syndrome (BWS), familial adenomatous polyposis (FAP), and Trisomy 18 (Edward’s Syndrome) have been well established [4]. Pediatric HCC, while also of hepatocytic origin, typically occurs de novo but can occur in the context of underlying hepatopathy or a predisposition syndromes [5,6]. HCC is consistently less chemosensitive, often presents at an advanced stage, and carries a far less favorable prognosis [3].

## 3. Epidemiology

HCC is a rare malignant neoplasm in the pediatric population, occurring with an incidence of 0.4 cases per million children per year in the U.S. in children 0 to 14 years of age and 1.4 cases per million children per year in patients between 15 and 19 years according to the Surveillance, Epidemiology, and End Result (SEER) database [3]. After HB, it is the second most common primary hepatic malignancy affecting pediatric, adolescent, and young adult (AYA) patients; however, it surpasses HB in accounting for 87% of malignant primary liver tumors in children aged 15–19 [6]. Geographical variation also accounts for global differences in the incidence of pediatric HCC. As in adult HCC, incidence is higher in areas with a high prevalence of viral hepatitis, though the impact of this predisposing factor has greatly diminished with widespread vaccination [7]. In the United States, the incidence of pediatric HCC has been stable over the past two decades [8]. In the pediatric and AYA population, HCC can be further classified as “de novo” (arising in the absence of known predisposing conditions or liver injury) or secondary HCC (arising due to known predisposing conditions or liver injury) [5]. Historically, fibrolamellar carcinoma (FLC) was considered a variant of conventional HCC, though the identification of a canonical chimeric fusion (DNAJB1-PRKACA) in FLC as well as differences in histology and clinical presentation have led to the acceptance of FLC as a distinct entity [9,10]. Inclusive of FLC, conventional HCC comprises the majority of HCC cases (73%) as compared with FLC, which comprises approximately 25% of cases [3]; however, this is with the recognition that FLC is likely underdiagnosed or inadvertently diagnosed as HCC.

## 4. Pathophysiology

### 4.1. Underlying Liver Conditions

Pediatric HCC is well known to occur in the setting of a broad array of predisposing conditions, whether secondary to underlying liver damage or a genetic predisposition to carcinogenesis. Two of the most common risk factors are chronic hepatitis (B or C) and hereditary tyrosinemia [1]. In the United States, a single-institution review of 61 cases reported an incidence of predisposing conditions in 53% of patients [1]. In this report, FLC cases were not excluded from this analysis and comprised a sizeable proportion of cases classified as de novo HCC. The most frequently identified predisposing conditions included cryptogenic cirrhosis/fibrosis, macrovesicular steatosis, Alagille syndrome, and progressive familial intrahepatic cholestasis (PFIC). This study also identified that children with predisposing conditions were more likely to present at a younger age and with smaller tumors, and had an increased median overall survival compared to de novo HCC cases, potentially secondary to surveillance in patients with conditions known to predispose to HCC. Here we review the most prevalent predisposing conditions and discuss considerations regarding pathogenesis, surveillance, and prevention.

#### 4.1.1. Chronic Viral Hepatitis

Globally, a prevalent cause of HCC in adult patients and some pediatric patients is chronic viral infection with hepatitis B or C, though vaccination has dramatically reduced the incidence of HCC caused by these pathogens [11]. Wen et al. reported an incidence of 32 cases of HCC per 100,000 person-years in a prospective study of Taiwanese children with chronic hepatitis B infection, and reported early “e” antigen seroconversion and cirrhosis as risk factors for HCC development [12]. A group from Japan confirmed the findings of early “e” seroconversion and cirrhosis as risk factors, in addition to demonstrating the utility of AFP monitoring for disease [13]. In young patients whose livers are growing rapidly, the hepatitis B virus can integrate into host DNA and progress to neoplasia quickly [14,15]. In contrast, hepatitis B acquired after the perinatal period progresses to cirrhosis and HCC more indolently, often taking up to 20 years. Regarding tumor drivers in viral hepatitis-associated HCC, Kim et al. identified unique differences between pediatric and adult cases, with lower expression of cyclin 1 and more frequent LOH at 13q in the pediatric cohort [16].

In contrast to adult HCC patients, the incidence of hepatitis C in children is quite low and infrequently leads to the development of HCC in the pediatric age group [17,18]. Generally, hepatitis C virus appears to mediate neoplastic transformation via a more prolonged time course and a distinct mechanism, acting via chronic inflammation resulting in cell turnover that leads to acquisition of somatic mutations that eventually progress to HCC.

#### 4.1.2. Metabolic Conditions

HCC is the most frequently diagnosed liver malignancy in children with inherited metabolic disorders. The classic paradigm for HCC pathogenesis in this context includes accumulation of toxic metabolites and chronic injury due to underlying disease impacting a critical role in metabolism [19]. Most of these conditions progress to cirrhosis before neoplasia, but this is not seen in all cases. The most frequently identified metabolic condition in cases of pediatric HCC is hereditary tyrosinemia, present in 18–37% of cases diagnosed before 18 months old [19]. Due to a deficiency of fumarylacetoacetate hydrolase, the terminal enzyme in tyrosine degradation, a buildup of earlier precursors is thought to mediate HCC development, as these precursors have been shown to have mutagenic capacity and direct inhibitory effects on DNA repair mechanisms [20,21]. NTBC, an organic herbicide that inhibits upstream tyrosine catabolism, reduces the incidence of acute symptomatic tyrosinemia and decreases the risk of HCC development when initiated in patients < 2 years of age [19]. Other inborn metabolic conditions that predispose to pediatric HCC include alpha-1 antitrypsin deficiency, hereditary hemochromatosis, glycogen storage disease (types I, III, IV), Wilson’s disease, and mitochondrial disorders [22]. Acquired metabolic syndromes, such as obesity, dyslipidemia, insulin resistance, and type 2 diabetes mellitus, have been implicated in the development of HCC in adults through the development of nonalcoholic fatty disease/steatohepatitis. The onset of many of these conditions occurs in the pediatric/adolescent–young adult (AYA) age group, suggesting a potential role for early lifestyle interventions, such as dietary modification, in the prevention of HCC.

#### 4.1.3. Other Cirrhotic and Vascular Disorders

There are multiple chronic cholestatic syndromes that increase the risk for HCC development, including long-term parental nutrition exposure, biliary atresia, Alagille syndrome, and progressive familial intrahepatic cholestasis (PFIC), in particular PFIC type 2 [23,24,25,26]. PFIC type 2 is characterized by a genetic defect in the bile acid transporter bile salt export pump (BSEP) encoded by ABCB11, and results in HCC via development of cirrhosis [27]. PFIC is managed by nutritional support, by supportive care for pruritis with ursodeoxycholic acid, and sometimes by biliary diversion procedures and liver transplantation. Exome sequencing of HCC cases associated with PFIC type 2 has revealed an absence of pathogenic somatic mutations and widespread copy number gain [28]. HCC is also known to occur in conditions that affect the vascular integrity of the liver, including congenital portosystemic shunts, venous outflow obstruction (Budd–Chiari syndrome), and as a late effect of the Fontan procedure [29].

### 4.2. Molecular Mechanisms

Pediatric HCC can occur de novo in an otherwise healthy liver or in the context of predisposing conditions, as described above. This biologic heterogeneity as well as the rarity of the condition has limited unifying conclusions drawn from molecular characterization of pediatric HCC compared to the adult disease. In adult HCC, the landscape of genomic alterations and the presence of two distinct molecular subgroups (proliferative class and non-proliferative class) have been defined [30,31]. Studies in adults have also revealed a high frequency of TP53 and TERT promoter mutations, as well as Wnt pathway activation occurring through missense mutations or intragenic deletions in CTNNB1 [32].

Importantly, understanding the interplay between predisposing conditions, environmental factors, and stochastic somatic events driving pediatric HCC is critical to understanding disease pathophysiology and improving upon available therapies. In an effort to address this gap, Haines et al. analyzed a cohort of 15 de novo HCC cases (non-fibrolamellar) using a cancer-associated gene panel, together with copy number and gene expression analyses [33]. This work revealed considerable heterogeneity in molecular alterations, and importantly established that tumors arising in children with underlying liver disease were molecularly distinct and lacking in detectable oncogenic drivers. Copy number analysis demonstrated focal heterozygous or homozygous copy number loss of CDKN2A in 4 and 2 of 15 patients, respectively, and partial loss of RB1 in 4 cases. Overall, 10 of 15 tumors had copy gain or loss of one or more whole chromosomal arms, but high-level segmental amplifications were not observed. Of note, tumors from patients with underlying liver disease had fewer copy number alterations compared to those with de novo HCC.

In this study, genomic sequencing for pathogenic molecular alterations revealed an average of 1.6 potentially pathogenic variants in cancer genes per sample, with no pathogenic mutations identified in seven cases (including all five cases with underlying liver disease) [33]. Wnt pathway alterations were identified in six (40%) cases, with deletions affecting exon 3 of CTNNB1 detected in four cases (27%). One case had an inversion in APC that produced two expressed in-frame fusions resulting in decreased expression of APC, which was ultimately shown to be a germline event accompanied by a second hit, an inactivating nonsense mutation in APC. RNA sequencing and hierarchical clustering supported that conventional HCC tumors were distinct from FLC tumors and normal liver. TERT (*n* = 2) and TP53 (*n* = 1) mutations were observed to occur less frequently than in adult cases. With the recognition that this study represents a small sample size, the authors suggest that a comprehensive molecular diagnostic approach is warranted for pediatric HCC. They also conclude that while pediatric HCC shares some oncogenic pathways with adult HCC and pediatric HB, generally these tumors carry more mutations than typically seen in HB. Of interest, the specific molecular alterations that result in Wnt activation, altered telomere maintenance, and MAPK/ERK activation seem to differ between pediatric and adult cases. Mosca et al. identified that the transcriptional factor homeobox gene, LHX2, appears to serve as a tumor suppressor in both HCC and hepatoblastoma in in vitro studies [34]. When down-regulated, LHX2’s inhibition of Wnt was impaired.

The spectrum of molecular alterations in pediatric liver tumors has been more broadly defined in two additional publications, one that sequenced 35 hepatoblastomas [35], and another that sequenced 154 hepatoblastomas and nine pediatric HCC cases [36]. The former study (Sumazin P, et al.) identified point or frameshift mutations in CTNNB1 in the majority (89%) of hepatoblastoma cases. Among the remaining (wildtype CTNNB1) cases, somatic alterations in NFE2L2, ARID1a, MLL2, and TERT were among the more frequently identified. TERT promoter mutations were observed in two older patients. Germline alterations in APC, MLL2, and ARID1a were identified in a subset of patients. High NFE2L2 activity and high expression of LIN28B, HMGA2, SALL4, and AFP were predictive of poor prognosis. The latter study, by Nagae et al., conducted genomic profiling, identifying similar genomic patterns, along with a low tumor mutational burden in classical hepatoblastoma (med 0.1 mut/Mb) compared with adult HCC (3.06 mut/Mb). Genome-wide methylation analyses revealed a subset of cases with aberrant hypermethylation of *DLX6-AS1* as a potential marker of poor 5-year EFS and poor prognosis.

### 4.3. Germline Predispositions

HCC does not commonly develop in patients with germline cancer predisposition syndromes; however, there are case reports of HCC affecting patients with familial adenomatous polyposis (FAP) and neurofibromatosis [3,37]. A publication from the adult literature offers further insights into cancer predisposition-related HCC risk from a prospective cohort of 217 patients with the disease who underwent multigene panel testing for 134 cancer predisposition genes [38]. Pathogenic/likely pathogenic (P/LP) germline variants were identified in 11.5% of patients, including BRCA2, MSH6, and PMS2, as well as FANCA and BRIP1. A separate retrospective analysis was also performed on 219 patients with HCC who underwent testing at a commercial laboratory, identifying 5.9% of patients with P/LP variants in genes including APC, BRCA1, BRCA2, MSH2, and TP53. Of note, a high proportion of these adult patients also had cirrhotic liver disease or viral hepatitis, so the causality of these variants remains unclear. Another group identified 2% of patients with HCC undergoing liver transplantation to have a pathogenic variant in established cancer predisposition genes, and they concluded that this rarity suggests that germline testing may not be warranted in the adult HCC population [39]. Studies of germline predisposition genes in pediatric HCC are limited.

## 5. Clinical Considerations

Children who are diagnosed with HCC frequently present with abdominal pain and distension, nausea or vomiting, decreased appetite, and weight loss. They are typically found to have an abdominal mass on physical examination. A provisional diagnosis is made based on presenting features: elevated alpha fetoprotein (seen in approximately 2/3 of cases), patient age, and the presence of hepatic lesions on ultrasound or cross-sectional imaging [3]. On imaging, pediatric de novo HCC often presents as a well-demarcated mass within a healthy liver; occasionally with accompanying satellite lesions or rarely as a more diffuse, infiltrative process. In contrast, HCC that arises in the context of underlying liver disease is more frequently multifocal and associated with background liver pathology such as cirrhosis. On histopathology, HCC cells are larger than HB cells and typically display nuclear pleomorphism, prominent nucleoli, and atypical mitoses. Pediatric HCC is graded into well-differentiated, moderately differentiated, and poorly differentiated lesions, which each have clearly defined characteristics [40]. By immunohistochemistry, HCC cells stain positive for HepPar1 and arginase; glycogen synthase and beta catenin staining are more variable.

Disease is likely to spread hematogenously to the lungs but can also involve lymph nodes or the peritoneum. Metastases to the brain are uncommonly seen in relapsed or refractory disease. The staging systems used in adult HCC, including TNM and the Barcelona staging system, are not readily used in pediatric patients. Instead, pediatric HCC cases are traditionally classified as resectable upfront, unresectable, and/or metastatic. The role of PRETEXT, while prognostic for HB, has not been rigorously studied for HCC. It is now being interrogated in an analysis by the Children’s Hepatic Tumor International Collaboration (CHIC) [41]. Other factors associated with an inferior prognosis include increasing age, multifocal disease, unresectable disease, and the presence of metastasis [42]. For curative potential, complete resection of the HCC tumor is imperative, but fewer than 20–30% of patients have resectable disease at the time of diagnosis [43]. As such, the overall prognosis for pediatric HCC is inferior to that for HB. For patients with upfront, resectable disease, outcomes in a small series have been reported as >80% [44]. Given the majority of patients present with advanced disease, overall outcomes are typically quite poor, in the range of 10–30%. Survival after transplantation is considerably improved, with one case series reporting an overall survival rate of 75% [45].

Significant efforts are focused on utilizing chemotherapy and studying novel therapies (see below) in the neoadjuvant setting to increase resectability. Unlike adult HCC for which responses to chemotherapy are rare, approximately 50% of children experience an objective response to neoadjuvant chemotherapy, which is posited to be due in part to the increased incidence of de novo disease in livers that are otherwise healthy and/or the fact that many HCC tumors may harbor elements of HB that respond better to chemotherapy [41]. Thus, the treatment of pediatric HCC has diverged from the approach pursued for adult patients. Historically, pediatric HCC patients were eligible for enrollment in pediatric cooperative group trials that grouped the treatment of patients with both HB and HCC. Treatment protocols used therefore included the following: cisplatin, 5-fluorouracil, and vincristine; cisplatin/doxorubicin; cisplatin, tetrahydropterynyl, and doxorubicin; cisplatin and pirarubicin; carboplatin/etoposide. The drugs incorporated were dependent upon the consortium running the trial. There are some clinical data to suggest that the combination of gemcitabine and oxaliplatin (GemOx) is active in pediatric patients with recurrent HCC, with 29% of patients achieving a partial response and another 29% with stable disease for 3–16 months in one study of 24 patients who received this therapy [46]. These data were used to inform the development of the unresectable/metastatic arms of the PHITT trial (see below). Table 1 summarizes the historic therapeutic approaches and outcomes reported by the international pediatric cooperative groups which previously studied HCC. Relapsed HCC is also under active study by a global pediatric consortium via the sourcing of retrospective relapse data [47].

Intra-arterial chemotherapy has also been studied in certain protocols, as has the role of autologous stem cell rescue. While response rates varied, generally consistently low overall survival rates were observed [41,48]. Increasing use of orthotopic liver transplantation (OLT) offers a curative surgical option in patients with advanced localized disease; however, there are currently no established consensus criteria for selecting which pediatric HCC patients should proceed to OLT, and patients with metastatic disease are uniformly excluded [3]. The recently concluded Pediatric Hepatic tumors International Therapeutic Trial (NCT03017326: PHITT/AHEP1531/JPLT-4), a collaborative effort between the Société Internationale d’Oncologie Pédiatrique-Epithelial Liver Tumours Group, the Children’s Oncology Group, and the Japanese Children’s Cancer Group, is the first to prospectively study a uniform treatment approach in localized versus unresectable/metastatic HCC. Analysis of the data from this trial may shed light on chemo-responsiveness in pediatric HCC. Importantly, this trial also includes correlative biology and aims to evaluate HCC tumor specimens with molecular characterization to validate newly identified molecular biomarkers correlating with known clinical prognostic factors and outcome. Locoregional treatment approaches, more commonly used in adults, such as trans-arterial chemotherapy (TACE) and more recently selective internal radiation therapy (SIRT) with Yttrium-90 microspheres, as well as stereotactic body radiotherapy (SBRT), have been used in both the palliative setting and as a bridge to transplant [50,51,52]. Such locoregional therapies were permissible in the unresectable/metastatic HCC arm of the PHITT trial but will require further study. A recent single-institution retrospective study also demonstrates safety and feasibility of percutaneous ultrasound-guided radiofrequency ablation in ten patients with relapsed HCC [53].

## 6. Novel Therapeutic Targets

Given the dismal overall and event-free survival with chemotherapeutic approaches to date, there has been significant interest and effort dedicated to expanding the repertoire of therapeutic options for children and adolescents with HCC (Figure 1).

The multi-tyrosine kinase (TKI) inhibitor sorafenib was the first agent to be approved for adult HCC, and its efficacy was assessed in seven children with unresectable tumors treated in combination with cisplatin/doxorubicin (PLADO) in Germany [30,49]. In a retrospective review of these patients, PLADO/sorafenib resulted in a partial response in four, stable disease in two, and progression in one. Three patients were alive after complete resection at the time of the report. There is a clear biological rationale for the use of sorafenib in HCC given that these tumors are known to be highly vascular. Additionally, adult HCC tumors frequently overexpress vascular endothelial growth factor (VEGF) and other factors associated with tyrosine kinase signaling pathways. There are also strong preclinical data to support sorafenib usage, including tumor growth inhibition in animal models of HCC [54]. Other multi-TKIs including lenvatinib and cabozantinib are under study in adult HCC [55]; lenvatinib is currently a mainstay of therapy for adults given evidence of non-inferiority and improved tolerability [56].

Another potential molecular target in a subset of HCC cases emerges from the work of Lanelli et al. that mapped the genomic alterations occurring in human and mouse HCCs induced by defects in hepatocyte biliary transporters, including PFIC type 2. This group demonstrated that copy number gains frequently involve the MAPK signaling pathway and in particular direct regulators of JNK (Jun N-terminal kinase). Pharmacological inhibition of JNK slowed progression in murine HCC models [28]. Given the established role of Wnt signaling, there is also a rationale to target this pathway [33]. Tegavivint, a compound which interferes with the binding of beta-catenin to TBL1, is currently being studied in relapsed/refractory solid tumors in the setting of a Phase I/II trial (NCT04851119).

There is also now established interest in immunotherapeutic approaches to targeting HCC, including immune checkpoint inhibition and cellular-based therapies for other novel cell surface targets [57,58,59]. Checkpoint inhibition with atezolizumab and bevacizumab has been established as an adult standard, with dual checkpoint inhibition as a close second, but these combinations have not yet been well studied in pediatric patients [60]. Currently, there is an open Phase II clinical trial testing the role of checkpoint inhibition with pembrolizumab in patients with HCC aged 0–30 (NCT04134559); this trial will soon be amended to study dual checkpoint inhibition.

Glypican 3 (GPC3) is a cell surface glycoprotein that has emerged as a potential target in HCC; accordingly, it was detected by IHC in all patients in the Haines et al. cohort, including patients with and without underlying liver conditions [33,61,62]. A novel IL-15 armored anti-GPC3 chimeric antigen receptor (CAR) has been in early phase clinical trials (NCT02932956), with anticipated launch of a dual-armored IL-15/IL-21 CAR trial to come (NCT06198296). GPC3 is also the target of a pediatric peptide vaccine trial which was conducted in Japan (UMN 000006357) and a pediatric monoclonal antibody trial underway in the United States (NCT04928677). Another open pediatric Phase I/II trial is studying the safety and efficacy of ET140203 (Eureka Therapeutics) in pediatric patients with hepatoblastoma, HCC, or HCN NOS; this construct uses a TCR mimic antibody to target an AFP peptide bound to HLA-A2 (NCT04634357).

## 7. HCN-NOS

One of the challenges to the classification of pediatric/AYA hepatic neoplasms, in addition to their relative rarity, is the lack of a consensus classification system within and between international consortia. To address this, in 2012 an international pathology symposium was conducted to develop an international consensus classification system, necessary for collaboration between international cooperative group consortia. As a result of this effort, a new provisional entity, hepatocellular malignant neoplasm NOS (HCN-NOS), was created to categorize a histologically diverse group of tumors for which a consensus classification could not be reached [63]. This entity overlaps with a group of tumors described initially in 2002 as transitional liver cell tumors (TLCTs) [64]. This intermediate entity occurs more frequently in older children and AYA patients, with histological and molecular features of both HB and HCC [63].

In 2022, a multi-institutional consortium, led by investigators at Baylor College of Medicine, sought to perform a molecular characterization of a clinically annotated set of HCN-NOS cases [65]. They found that a subset of HCN-NOS cases was characterized by higher mutation rates as compared with HB (but lower than seen in HCC) and more frequent copy number alterations, suggesting greater genomic instability as compared with HB. In addition, this subset of HCN-NOS cases was enriched for alterations and mutations in genes and pathways associated with stem cell pluripotency and in PI3K-AKT-mTOR signaling. The most commonly observed mutations were in the TERT promoter region and genes such as APC, KEAP1, NFE2L2, ARID1A, ARID1B, and PIK3CA. The patients in this cohort had generally poor outcomes. Patients who were transplanted had superior outcomes to those treated with chemotherapy and surgery. Given the inherent limitations of this small retrospective analysis, the authors suggest a larger-scale prospective trial offering uniform chemotherapy and surgical protocols to treat patients with HCN NOS as a distinct cohort. Patients with HCN-NOS are traditionally treated with HB-directed regimens, but their tumors often develop rapid resistance to these therapies; therefore, complete surgical resection remains the most important aspect of the management of these malignancies.

## 8. Conclusions

In conclusion, pediatric HCC remains a vastly understudied disease, associated with a poor prognosis, and an unmet need for individualized study. Recognition of the disease as a unique entity, on a spectrum with HB and HCN NOS, as well as organized efforts on an international stage to commit to a uniform approach to the histologic diagnosis, genomic profiling, and treatment approach will undoubtedly power conclusions from ongoing and future studies and enhance our understanding of this disease while hopefully improving outcomes.

## Figures and Tables

**Figure 1 ijms-26-01252-f001:**
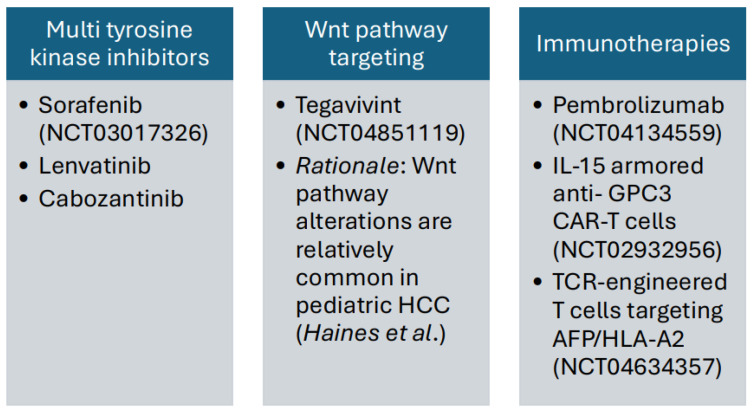
Key categories of novel therapeutic targets for pediatric HCC [33].

**Table 1 ijms-26-01252-t001:** Historical approaches in treating pediatric hepatocellular carcinoma and associated outcomes.

Study	Chemotherapy	Outcomes (5 Years OS)	
INT-0098 (CCG/POG)	C5V vs. CD	PRETEXT Stage: I III IV	(OS) (88%) (23%) (10%)
SIOPEL1	PLADO	PRETEXT Stage: I/II III IV M+	(OS) (44%) (22%) (8%) (9%)
SIOPEL 2 and 3	SuperPLADO	All Patients Primary Resection Delayed Resection Unresectable	(22%) (~50%) (~40%) (0%)
GPOH	PLADO + Sorafenib	PRETEXT Stage: II: III: IV:	CR (3, 12–27 mo[1OLT]), PD (1, 23 mo), DOD (1) CR (2, 18–32 mo[1OLT]), SD (1, 5 mo) CR (1, 12 mo), PD (1, 18 mo), DOD (2)

CCG: Children’s Cancer Group; POG: Pediatric Oncology Group; GPOH: German Pediatric Oncology Hematology Group; OS: overall survival; M+: Metastatic Disease Present; OLT: orthotopic liver transplant. Chemotherapy regimens: C5V: cisplatin + 5-fluorouracil (5FU) + vincristine; CD: cisplatin + doxorubicin (as dosed in INT0098); PLADO: cisplatin + doxorubicin (as dosed in SIOPEL); SuperPLADO: cisplatin + doxorubicin + carboplatin; CR: complete response; PD: progressive disease; SD: stable disease; DOD: dead of disease [40,48,49].

## Data Availability

No new data were created or analyzed in this study.

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
