# Peer review of "Pediatric Hepatocellular Carcinoma: A Review of Predisposing Conditions, Molecular Mechanisms, and Clinical Considerations"

_ijms, 2025, doi:10.3390/ijms26031252_

Round 1

Reviewer 1 Report

Comments and Suggestions for Authors

The manuscript titled "Pediatric Hepatocellular Carcinoma: A Review of Predisposing Conditions, Molecular Mechanisms, and Clinical Considerations" provides a comprehensive overview of the current state of multidisciplinary knowledge on paediatric hepatocellular carcinoma (HCC).  The work is comprehensive and well-structured, encompassing a range of aspects, including diagnostic procedures, prognosis and subtyping, as well as clinical management and therapeutic approaches. The manuscript is well written and incorporates a substantial amount of relevant literature. The manuscript may be suitable for publication if the authors address the following overall minor issues and suggestions.

- The authors state at the outset that approximately 10% of cases occur in the context of a predisposing hepatic or systemic disease. The number should be justified or the phrase rewritten, as the manuscript also states that this value is highly dependent on the regional context and can reach up to 40% (e.g. hepatitis) or even be genetic predisposition alone, asthe percentage of genetically predisposed cases exceeding 10% has been observed in some studies.

- The authors do not cite works published beyond 2022. However, between 2023 and 2024, approximately a dozen original studies emerged, some of which are highly pertinent to the current study. While the authors are undoubtedly at liberty to select the references for this non-systematic review, it would be advisable to incorporate more recent works in order to enhance the impact of this manuscript.

- In the lines 163-164, the authors cite an incorrect statement in reference 27 concerning elevated levels of DKK1 consistent with Wnt pathway activation. DKK1 is a well-known inhibitor of Wnt signalling. While the original reference is clearly erroneous, the effect of DKK1 increase is cancelled out by mutation activation downstream of Wnt, making this a relatively minor issue.

- Given the potentially broader scope of readers of such an article, it would be useful to make a separate small section dedicated to highlighting the differences between HB and HCC. The subject is touched upon by the authors on several occasions, but a separate section would be more appropriate.   

- It would be beneficial to introduce the complete meaning of the rare acronyms FLC and HCN-NOS at least once, potentially in the section title for HCN-NOS.  

Author Response

Reviewers Comment 1: "The authors state at the outset that approximately 10% of cases occur in the context of a predisposing hepatic or systemic disease. The number should be justified or the phrase rewritten, as the manuscript also states that this value is highly dependent on the regional context and can reach up to 40% (e.g. hepatitis) or even be genetic predisposition alone, asthe percentage of genetically predisposed cases exceeding 10% has been observed in some studies."

Author's Response to Comment 1: Thank you very much for taking the time to review this manuscript. Please find the detailed responses below and the corresponding revisions/corrections highlighted/in track changes in the re-submitted file The authors have added two references and adjusted the text to state: "While the majority of HCC cases are sporadic, arising de novo in a structurally and functionally normal liver, predisposing anatomic, metabolic, or genetic conditions have been reported to occur in as many as 40-53% of cases (1, 2)"

Reviewers Comment 2:"The authors do not cite works published beyond 2022. However, between 2023 and 2024, approximately a dozen original studies emerged, some of which are highly pertinent to the current study. While the authors are undoubtedly at liberty to select the references for this non-systematic review, it would be advisable to incorporate more recent works in order to enhance the impact of this manuscript"

Author's Response to Comment 2: Thank you for this comment which is highly relevant. and an oversight on our part.

Reviewers Comment 3: "The authors do not cite works published beyond 2022. However, between 2023 and 2024, approximately a dozen original studies emerged, some of which are highly pertinent to the current study. While the authors are undoubtedly at liberty to select the references for this non-systematic review, it would be advisable to incorporate more recent works in order to enhance the impact of this manuscript"

Author's Response to Comment 3: Thank you for noting this important omission. The authors have now added two more relevant references from 2023 and 2024.

Reviewer's Comment 4: "- In the lines 163-164, the authors cite an incorrect statement in reference 27 concerning elevated levels of DKK1 consistent with Wnt pathway activation. DKK1 is a well-known inhibitor of Wnt signalling. While the original reference is clearly erroneous, the effect of DKK1 increase is cancelled out by mutation activation downstream of Wnt, making this a relatively minor issue."

Author's Response to Comment 4: The authors apologize for this error. The reference to elevated DKK levels has been removed from the text of the article.

Reviewer's Comment 5: "Given the potentially broader scope of readers of such an article, it would be useful to make a separate small section dedicated to highlighting the differences between HB and HCC. The subject is touched upon by the authors on several occasions, but a separate section would be more appropriate."

Author's Response to Comment 5: The authors have added a separate section which provides a concise comparison of the two diagnoses.

Reviewer's Comment 5: "It would be beneficial to introduce the complete meaning of the rare acronyms FLC and HCN-NOS at least once, potentially in the section title for HCN-NOS."

Author's Response to Comment 5: The authors apologize for this oversight and have added the expanded meaning of these acronyms.

Reviewer 2 Report

Comments and Suggestions for Authors

A well-prepared review work on HCC. It concerns contemporary views on the pathomechanism (including molecular) of the development of HCC and new methods of treating this disease. Basically, I have no comments on the content, as it presents a modern view of the disease. I would suggest pointing out in the introduction that the disease affects older children(v. 25). Please expand on some abbreviations when using them for the first time, such as: HCN-NOS, FLC, PHITT (trail).

The references are contemporary and well selected.

It would be a great help to present metabolic/molecular/ diagnostic and therapeutic directions in the form of a table.

Author Response

Reviewer Comment 1: "A well-prepared review work on HCC. It concerns contemporary views on the pathomechanism (including molecular) of the development of HCC and new methods of treating this disease. Basically, I have no comments on the content, as it presents a modern view of the disease. I would suggest pointing out in the introduction that the disease affects older children(v. 25). Please expand on some abbreviations when using them for the first time, such as: HCN-NOS, FLC, PHITT (trail). 

Author's Comment: The author's appreciate the reviewer's very positive comments. The requested abbreviations have been added.

Reviewer Comment 2: 

ors

A well-prepared review work on HCC. It concerns contemporary views on the pathomechanism (including molecular) of the development of HCC and new methods of treating this disease. Basically, I have no comments on the content, as it presents a modern view of the disease. I would suggest pointing out in the introduction that the disease affects older children(v. 25). Please expand on some abbreviations when using them for the first time, such as: HCN-NOS, FLC, PHITT (trail).

The references are contemporary and well selected.

It would be a great help to present metabolic/molecular/ diagnostic and therapeutic directions in the form of a table.

Authors Response to Comment: The authors thank the reviewer for their kind review. A table showing therapeutic approaches to HCC has been added